# A Novel Chaperone-Based Cancer Vaccination Enhances Immunotherapeutic Responsiveness Through T Cell Amplification and Tumor Immune Remodeling

**DOI:** 10.3390/vaccines13111096

**Published:** 2025-10-25

**Authors:** Xueqian Cheng, Zheng Liu, Jinyang Cai, Xiaofei Yu, Chunqing Guo, Wenjie Liu, Masoud H. Manjili, Syed A. Shah, Elizabeth A. Repasky, John R. Subjeck, Xiangyang Wang

**Affiliations:** 1Department of Cellular, Molecular and Genetic Medicine, Virginia Commonwealth University School of Medicine, Richmond, VA 23298, USA; 2Department of Microbiology and Immunology, Virginia Commonwealth University School of Medicine, Richmond, VA 23298, USA; 3Massey Comprehensive Cancer Center, Virginia Commonwealth University School of Medicine, Richmond, VA 23298, USA; 4Richmond Veterans’ Affairs Medical Center, Richmond, VA 23249, USA; 5Department of Surgery, Virginia Commonwealth University School of Medicine, Richmond, VA 23298, USA; 6Roswell Park Comprehensive Cancer Center, Buffalo, NY 14203, USA; 7Institute of Molecular Medicine, Virginia Commonwealth University School of Medicine, Richmond, VA 23298, USA

**Keywords:** vaccine, chaperone, tumor microenvironment, immune checkpoint inhibitor

## Abstract

Background/Objectives: Preclinical and clinical evidence supports a chaperone-based vaccination platform for cancer immunotherapy. The objective of this study is to interrogate the next generation of chaperone-based immune modulator, termed Flagrp170, which was constructed by fusing a defined NF-κB-activating microbial sequence with a large stress protein with a superior antigen-holding/presenting property in the setting of antigen-targeted cancer vaccination. Methods: Bone marrow-derived dendritic cells were treated with Flagrp170 protein or an unmodified parental chaperone molecule (i.e., Grp170), followed by an analysis of DC activation and DC-mediated T cell priming using both in vitro and in vivo models. Antitumor vaccine responses in mice receiving tumor antigens (e.g., gp100, Her2/neu) complexed with Flagrp170 or Grp170 were examined through multiple immune assays. The potential use of a Flagrp170-based chaperone vaccine to sensitize tumors to anti-PD-1 therapy was also evaluated. Results: Flagrp170 not only retains the intrinsic ability of the parent chaperone to facilitate antigen cross-presentation, but also acquires a unique capacity to stimulate DCs efficiently through the engagement of TLR5-NF-κB signaling. This chimeric chaperone shows superior activity compared to the unmodified parental molecule, resulting in enhanced DC activation and T cell priming. Vaccination with Flagrp170 complexed to tumor antigens induces a robust T cell response against primary tumors and metastases, a process critically dependent on CD8^+^ DCs. Additionally, the Flagrp170 chaperone vaccine can efficiently generate and expand tumor-reactive T cells. The consequent remodeling of the tumor microenvironment towards a Th1/Tc1 dominant immune phenotype significantly potentiates cancer responsiveness to anti-PD1 therapy. Conclusions: Given the safety and T cell stimulation profiles of the chaperone–antigen complex vaccine already established in our recent clinical trial, this new generation of chaperone cargo, capable of delivering both antigenic targets and pathogen-associated immunoactivating signals simultaneously, represents a promising strategy to potentially improve the low response rates in patients receiving immune checkpoint inhibitors.

## 1. Introduction

Immunotherapy has been established as the standard-of-care treatment option for metastatic and locally advanced cancers [1,2]. Immune checkpoint blockade therapies, which target inhibitory immune regulatory proteins such as PD-1/L1 and cytotoxic T lymphocyte antigen 4 (CTLA-4), have demonstrated remarkable clinical responses in the treatment of several cancer types, including melanoma [3,4,5]. However, the majority of patients do not respond well to these immune checkpoint inhibitors (ICIs), indicating a need for innovative approaches to enhance response rates. Both preclinical and clinical evidence suggests that a poor response to ICIs may be due to a low cancer mutational burden, poor tumor immunogenicity, or the absence of pre-existing tumor-reactive CD8^+^ T cells [6,7]. Clinical studies have shown that ICI-responsive cancers tend to express inflammatory or IFN-γ signature genes and a T cell inflamed phenotype [8,9,10,11]. Indeed, patients with T cell-rich tumors (i.e., T cell inflamed or immunologically ‘hot’) often experience longer progression-free and overall survival [12]. Thus, the use of a novel cancer vaccination to strategically boost pre-existing or generate de novo immune responses, especially in tumor sites, may provide opportunities for improved treatment outcomes with ICIs.

A large body of evidence supports the use of evolutionarily conserved chaperone molecules to develop cancer vaccines, mainly due to their superior capacity in shuttling antigens preferentially to antigen-presenting cells (APCs) via an interaction with their specific receptors for highly efficient cross-presentation [13]. We have developed synthetic chaperone vaccines by complexing large stress proteins, e.g., heat shock protein 110 (Hsp110) or glucose-regulated protein 170 (Grp170), non-covalently with clinically relevant tumor protein antigens (e.g., gp100, Her2/neu) and demonstrated their superior immunostimulatory activities for inducing antigen-specific CD8^+^ cytotoxic T lymphocyte (CTL) responses to cancers [14,15]. Our phase I trial showed that the recombinant human Hsp110-gp100 chaperone vaccine could activate CD8^+^ T cells in patients with advanced, pretreated melanoma [16], which supports therapeutic applications using these antigen-targeted chaperone complex vaccines to generate and/or expand tumor-reactive T cells for cancer eradication.

To further strengthen the immunostimulatory activity of the chaperone-based vaccine platform, we have created a chimeric molecule by incorporating a defined NF-κB-stimulating sequence derived from microbial flagellin protein, a ligand for toll-like receptor (TLR5), to the backbone of parental chaperone molecule Grp170 [17]. This new generation of chaperone-based immune modulator, termed Flagrp170, can promote antigen cross-presentation while concurrently activating dendritic cell (DC) function by providing an immunostimulatory microbial signal [17]. While the immunotherapeutic potency of Flagrp170 upon adenovirus delivery to the tumor lesions has been recently reported [17,18], its potential use as a protein antigen carrier in the setting of cancer vaccination has not been examined. In this study, we evaluate this new class of immune modulator when complexing with tumor-associated protein antigen (e.g., gp100) for the generation of tumor-reactive T cells. Additionally, we determine the feasibility of using this chimeric chaperone–antigen complex vaccine to potentiate cancer responsiveness to ICIs through the expansion of antigen-specific T cells and reprogramming of tumor immune niches.

## 2. Materials and Methods

### 2.1. Mice and Cell Lines

C57BL/6 mice, BALB/c mice, FVBN202 mice, Tlr5^−/−^ mice, Batf3^−/−^ mice, and Pmel transgenic mice were purchased from the Jackson Laboratory (Bar Harbor, ME, USA). All experiments and procedures involving mice were approved by the institutional Animal Care and Use Committee (IACUC) of Virginia Commonwealth University. Human gp100-expressing mouse melanoma B16 cell line was from Dr. Alexander Rakhmilevich (University of Wisconsin-Madison, Madison, WI, USA). The rat Her2/neu-positive MMC tumor cell line, purchased from Applied Biological Materials Inc, was established from spontaneous mammary carcinoma of FVBN202 transgenic mice [19,20]. B16 and MMC cells were maintained in Dulbecco’s Modified Eagle’s Medium and RPMI1640 medium, respectively, supplemented with 10% fetal bovine serum (FBS), 10 mM 2-[4-(2-Hydroxyethyl)-1-piperazinyl]-ethanesulfonic acid (HEPES), 2 mM L-glutamine, 100 U/mL penicillin, and 100 µg/mL streptomycin. All cell lines were routinely tested for mycoplasma contamination using a PCR-based mycoplasma detection kit (ATCC, Manassas, VA, USA).

### 2.2. Reagents and Antibodies

Fluorochrome-conjugated anti-mouse monoclonal antibodies, including FITC-CD3 (17A2), APC-CD4 (GK1.5), PerCP/Cy5.5-CD8 (2.43), PerCP/Cy5.5-CD90.1 (Thy1.1), PE-IFN-γ (XMG1.2), PerCP/Cy5.5-CD11b (M1/70), APC-CD11c (N418), FITC-CD80 (16-10A1), PE-CD86 (GL1), PE-CD40 (3/23), and BV711-MHCII (IA/IE) (M5/114), as well as CD16/CD32 (2.4G2), isotype control rat IgG2B (RTK4530), and IgG1 (RTK2071), were purchased from Biolegend (San Diego, CA, USA). Human gp100_25–33_ (KVPRNQDWL) peptides and TRP2_175–192_ peptides (QIANCSVYDFFVWLHYYA) were from AnaSpec (Fremont, CA, USA). Antibodies against phosphor-IκBα (Ser32/36), IκBα, Phospho-p65, and p65 were purchased from Cell Signaling Biotechnology (Beverly, MA, USA). Mouse tumor necrosis factor (TNF)-α, interleukin (IL)-6, IL-12p40, IL-12p70, interferon (IFN)-γ, IL-1β, and IL-2 ELISA kits were purchased from Biolegend. Anti-PD-1 antibody (RMP1-14) was purchased from BioxCell (Lebanon, NH, USA). Mouse recombinant Grp170 and Flagrp170 were expressed in Sf21 insect cells using the BacPAK baculovirous expression system (Clontech, Palo Alto, CA, USA) as described previously [14,21]. Proteins were purified using nickel-chelating resins which were purchased from Qiagen (Valencia, CA, USA) and dialyzed against endotoxin-free PBS. Endotoxin levels in the recombinant protein preparations are approximately 10 EU/mg protein.

### 2.3. Dendritic Cell Culture and T Cell Priming In Vitro

Bone marrow-derived dendritic cells (DCs) were generated by culturing the mouse bone marrow cells in the presence of mouse GM-CSF from PeproTech (Rocky Hill, NJ, USA) [17]. Chaperone–antigen protein complexes were prepared as previously described [14]. Briefly, recombinant gp100 protein was incubated with Grp170 or Flagrp170 protein (1:1 molar ratio) at 45 °C for 30 min, followed by incubation at 37 °C for an additional 30 min. DCs were then pulsed with chaperone–antigen complexes (30 µg/mL) for 4 h and washed prior to co-culture with Pmel mouse-derived CD8^+^ T cells (1:5 ratio) for 72 h in 200 µL RPMI 1640 medium containing BrdU (10 µM/mL) in a round-bottom 96-well microtiter plate. T cell proliferation and activation was assessed by ELISA and flow cytometry analysis.

### 2.4. Adoptive T Cell Transfer

2 × 10^6^ T cells from Pmel mice were transferred into recipient mice by tail vein injection, followed by subcutaneous immunization with chaperone protein complexes next day. Spleen and lymph nodes were harvested 3 days later and subjected to flow cytometry analysis following antibody staining for CD90.1 and CD8. IFN-γ production by T cells was assessed by ELISA or intracellular cytokine staining.

### 2.5. Intracellular Cytokine Staining

Cells were stimulated by phorbol 12′-myristate-13′-acetate (PMA, 10 nM), ionomycin (0.5 µm), plus brefeldin A (BFA, 5 µg/mL) for 4 h. Cells were incubated with anti-CD16/CD32 antibodies for 20 min on ice, followed by staining with antibodies for surface markers. Cells were then fixed, permeabilized, and stained with PE-conjugated anti-IFN-γ antibodies for 30 min at room temperature prior to flow cytometry analysis.

### 2.6. Real-Time Quantitative PCR

Total RNA was extracted using TRIzol reagent which was purchased from ThermoFisher Scientific (Waltham, MA, USA). Quantitative Real-Time PCR (qRT-PCR) was performed using a carboxyfluorescein (FAM)-labeled probe set from ThermoFisher Scientific. Gene expression was quantified relative to β-actin expression and normalized to control group measurements using the standard 2^−ΔΔCt^ calculation method.

### 2.7. Tumor Studies

Tumors were established in the dorsal flank through the s.c. inoculation of 2 × 10^5^ of B16 or 2 × 10^6^ MMC cells in C57BL/6 or FVBN202 mice on day 0, respectively. Experimental lung metastases were established by iv. injection of 1 × 10^5^ of B16 cells in C57BL/6 mice. Mice were randomized into different treatment groups (*n* = 5) and received chaperone vaccines (50 µg, s.c) or anti-PD-1 antibodies (200 μg, i.p.) on days 5, 8, and 11. Tumor volume (mm^3^) was calculated using the formula V = (shortest diameter × shortest diameter × longest diameter)/2. Mice were euthanized when the tumor diameters reached 16 mm or caused a 10% weight loss in the mice. Tumors were collected for qRT-PCR analysis. Tumor tissues were processed with Miltenyi Biotec gentleMACS Octo Dissociator to prepare single cell suspensions.

### 2.8. Statistical Analysis

Data were analyzed using GraphPad Prism software (Version 10) and expressed as mean ± SD values. Statistical significance between groups within experiments was determined by Student’s *t*-test, the two-way ANOVA test, and the log-rank test. The data points were biological replicates. Data representative of three independent experiments are shown. A *p* value less than 0.05 was considered statistically significant.

## 3. Results

### 3.1. Flagrp170 Protein Enhances the NF-κB Signaling-Mediated Activation of DCs

To examine the immunostimulatory activity of the chimeric Flagrp170 molecule, we first prepared the recombinant Flagrp170 protein using a baculovirus–insect cell expression system [14,21]. The integrity and purity of the protein was confirmed through the Coomassie staining and immunoblotting (Figure 1A, Appendix A. original images and relative signal intensity A). Similarly to other molecular chaperones, Flagrp170 also showed preferential binding for professional APCs, e.g., DCs (Appendix A). When compared to bone marrow-derived DCs treated with parental unmodified Grp170, DCs treated with the newly created chimeric chaperone Flagrp170 displayed higher expressions of antigen-presenting molecule MHC II, co-stimulation molecule CD86, and chemokine receptor CCR7, which can facilitate the migration of activated DCs (Figure 1B) [22]. Additionally, Flagrp170 protein was more efficient than Grp170 in stimulating the production of pro-inflammatory or immunostimulatory cytokines by DCs, including IL-6, IL-12p40, TNF-α, and IL-1β (Figure 1C). Consistent with the ELISA results, qRT-PCR showed that gene transcription of *il6*, *il12a*, *tnfa*, and *il1b* were also sharply increased upon Flagrp170 treatment (Figure 1D). Since the chimeric Flagrp170 contains a defined sequence for NF-κB activation that is critical for the immunostimulatory function of DCs [23,24,25], we next examined its NF-κB-stimulating activity upon interaction with DCs. Immunoblotting analysis revealed that Flagrp170 protein efficiently induced an activation of the NF-κB signaling pathway, evidenced by the increased phosphorylation of P65 as well as the increased degradation of IκBα, associated with its phosphorylation (Figure 1E, Appendix A. original images and relative signal intensity B). We further determined the NF-κB-stimulating activity of Flagrp170 using dual luciferase reporter assays. HEK293 cells stably expressing TLR5 were transfected with an NF-κB-firefly luciferase reporter plasmid and a constitutively active thymidine kinase–renilla luciferase reporter plasmid, followed by stimulation with Grp170 or Flagrp170. We showed that Flagrp170, not unmodified Grp170, was able to activate NF-κB, as indicated by the significant elevation of luciferase activity (Figure 1F).

Collectively, our data suggest that the integration of the flagellin-derived microbial sequence confers Flagrp170’s superior capacity for DC activation by engaging the NF-κB signaling.

### 3.2. Flagrp170–Antigen Protein Complex Enhances DC-Mediated T Cell Priming

To test the ability of Flagrp170 to enhance the cross-presentation of tumor antigens in the chaperone complex, we carried out in vitro T cell priming assays. DCs were pulsed with recombinant gp100 protein, the Grp170-gp100 protein complex or Flagrp170-gp100 protein complex, followed by a co-culture with gp100-specifc CD90.1^+^CD8^+^ naïve T cells (i.e., Pmel cells) purified from Pmel17 TCR transgenic mice [26]. DCs loaded with the Flagrp170-gp100 complex resulted in significantly more IL-2 and IFN-γ production by Pmel T cells as compared with those treated with the Grp170-gp100 complex (Figure 2A). This was also confirmed by the increased BrdU incorporation into proliferating CD90.1^+^ T cells stimulated by Flagrp170-gp100 complex-pulsed DCs (Figure 2B), suggesting an enhanced T cell proliferation following the treatment of DCs with Flagrp170.

We further examined the immunogenicity of the Flagrp170–antigen chaperone complex in vivo using an adoptive T cell transfer model. gp100-specific CD90.1^+^ Pmel T cells were transferred to recipient mice, followed by a subsequent immunization with either the Grp170-gp100 complex or Flagrp170-gp100 complex. We showed that CD90.1^+^CD8^+^ Pmel cells in spleens (Figure 2C) and lymph nodes (Appendix A) displayed a more robust expansion in mice receiving the Flagrp170-gp100 complex when compared with those receiving the Grp170-gp100 complex. Additionally, intracellular cytokine staining revealed that Flagrp170-gp100 complex-immunized mice showed a marked increase in IFN-γ-producing CD90.1^+^CD8^+^ T cells compared with those immunized with the Grp170-gp100 complex (Figure 2C). The enhanced activation of CD8^+^ T cell response was also observed when Flagrp170 complexed with breast cancer antigen Her2/neu (Figure 2D), further supporting the increased capacity of Flagrp170 in antigen cross-presentation and T cell stimulation.

### 3.3. Flagrp170 Induces TLR5-Dependent DC Activation

Given that flagellin serves as a pathogen-associated molecular pattern recognized by TLR5 [27,28], we examined the potential involvement of TLR5 in the Flagrp170-enhanced activation of DCs. Upon the stimulation of wild-type (WT) or TLR5^−/−^ DCs with Flagrp170 protein, we showed that the loss of TLR5 abolished the ability of Flagrp170 to induce the phosphorylation of P65 and IκBα (Figure 3A, Appendix A. original images and relative signal intensity C). Additionally, Flagrp170-stimulated TLR5^−/−^ DCs produced significantly less cytokines (i.e., IL-12p40, IL-6) when compared with WT counterparts (Figure 3B), which was associated with the significantly reduced transcription of these cytokine genes in TLR5^−/−^ DCs (Figure 3C). Given that the NLR family CARD domain-containing protein 4 (NLRC4) inflammasome also recognizes intracellular flagellin [29], we also examined the potential involvement of NLRC4 in Flagrp170-enhanced DC activation. We showed that the lack of NLRC4 in DCs did not appear to have an effect on Flagrp170-stimulated cytokine production (Appendix A), supporting TLR5 as a major mediator of Flagrp170-enhanced DC activation. To determine the TLR5 dependence of Flagrp170 activity in the setting of antigen cross-presentation, we loaded WT or TLR5^−/−^ DCs with the Flagrp170-gp100 complex, and co-cultured them with gp100-specific CD8^+^ Pmel T cells. It was shown that the TLR5 deficiency significantly reduced the capability of DCs to stimulate the activation of Pmel T cells, evidenced by the decreased levels of cytokines IL-2 and IFN-γ present in the supernatant (Figure 3D). These results suggest that TLR5 on DCs is necessary for Flagrp170-enhanced DC activation and its subsequent T cell-stimulating function.

### 3.4. Flagrp170–Antigen Complex Vaccine Enhances Tumor Inhibition by Amplifying Tumor-Reactive T Cells for Remodeling of Tumor Immune Niches

We next examined the therapeutic activity of the Flagrp170-gp100 complex as a cancer vaccine. C57BL/6 mice established with B16 melanoma were immunized with the Flagrp170-gp100 complex, Grp170-gp100 complex, or were left untreated. We showed that treatment with the Flagrp170-gp100 vaccine resulted in the more effective inhibition of tumor growth compared to treatment with the Grp170-gp100 vaccine (Figure 4A). The qRT-PCR analysis of tumor tissues revealed that Flagrp170-gp100 immunization significantly elevated Th1 immunity-related cytokine genes, including *ifng* and *il12a* (Figure 4B). The FACS analysis further showed that Flagrp170-gp100 vaccination resulted in an increased tumor infiltration by both CD4^+^ cells and CD8^+^ cells (Figure 4C), particularly IFNγ-expressing CD8^+^ T cells (Appendix A), when compared with Grp170-gp100 vaccination, suggesting an improved ability of this chimeric Flagrp170 molecule to mobilize adaptive immune cells infiltrating and remodeling the tumor compartment. Similar observations were also made when luciferase was used as a model antigen for complexing with Flagrp170 in the treatment of mice with luciferase-expressing B16 tumors (Appendix A). Additionally, we showed that splenocytes or tumor-draining lymph node cells from Flagrp170-gp100 vaccinated mice produced significantly higher levels of INF-γ upon stimulation with gp100_25–33_ peptide than those from the Flagrp170-gp100 vaccinated mice (Figure 4D), suggesting a superior Flagrp170 activity in inducing a systemic antigen-specific T cell response.

To test whether Flagrp170 can improve the therapeutic outcome of vaccination against cancer metastases, we established experimental lung metastases through the *i.v.* injection of B16 tumor cells, followed by treatment with a Flagrp170 or Grp170-based chaperone vaccine. Vaccination with the Flagrp170-gp100 complex resulted in a more effective inhibition of lung metastasis compared with Grp170-gp100 vaccination, as shown by colony-forming assays using cell suspensions prepared from lung tissues (Figure 4E). Consistent with this observation, the Flagrp170-gp100 treatment induced the highest levels of gene transcription for *il15* and *il12a* in the lungs (Figure 4F), which have been linked to DC activation and are crucial for the induction and maintenance of a Th1-polarized antitumor CTL response. Indeed, the Flagrp170-gp100 enhanced metastatic inhibition correlated with the activation of CD8^+^ T cells in the metastatic lesions, indicated by the elevation of lung-infiltrating, IFN-γ-producing CD8^+^ T cells (Appendix A).

To further confirm the enhanced antitumor efficacy of the Flagrp170-based chaperone vaccine, we immunized mice carrying Her2/neu-expressing mammary tumors with the Flagrp170-Her2/neu complex, Grp170-Her2/neu complex, or left them untreated. Similarly, the vaccination with Flagrp170-Her2/neu resulted in a more effective inhibition of tumor growth compared with the Grp170-Her2/neu vaccination (Figure 4G). The qRT-PCR analysis of tumor tissues showed that the Flagrp170-Her/neu vaccination was more efficient than the conventional Grp170 chaperone vaccine in upregulating the immune genes *ifng* and *granzyme B* (Figure 4H). Additionally, the *foxp3* gene in tumor sites was significantly elevated following Flagrp170-Her/neu vaccination (Appendix A), suggesting that an immune homeostatic mechanism may be engaged. The expression of immune checkpoint molecules *pd1* and *pdl2* was also significantly upregulated following Flagrp170-Her/neu vaccination (Figure 4H), further supporting the enhanced activity of Flagrp170 in immunologically reprogramming the tumor microenvironment. The splenocytes or lymph node cells from the Flagrp170-Her2/neu vaccination group also produced higher levels of IFN-γ after stimulation with MMC tumor lysates or Her2/neu protein (Figure 4I). Those results together support the superior immunostimulatory activity of Flagrp170, as well as the potential use of a Flagrp170-based chaperone vaccine to expand tumor-reactive T cells for reprogramming tumor immune niches.

### 3.5. CD8^+^ DCs Are Required for Therapeutic Efficacy of Flagrp170 Chaperone Vaccine

CD8^+^ DCs are uniquely efficient at capturing pathogen or tumor antigens and presenting them on their MHC class I molecules to activate CD8^+^ T cells [30]. Our initial ex vivo study showed that Flagrp170 was significantly more efficient than unmodified Grp170 in stimulating the maturation and activation of CD8^+^ DCs upon administration to mice (Appendix A flow cytometry gating strategy for CD8^+^CD11c^+^DCs). To further validate this observation, we injected Grp170 or Flagrp170 protein into the bilateral sides of the same mouse, respectively, followed by the recovery of draining lymph nodes from each immunization site to determine phenotypic changes in CD8^+^ DCs. We showed the enhanced activation of CD8^+^CD11c^+^ DCs in the draining lymph nodes from the Flagrp170 injection site in comparison with those DCs in the draining lymph nodes from the contralateral Grp170 injection site, indicated by the elevated expression of CD86, CD40, and MHCII molecules (Figure 5A).

We next used Batf3^−/−^ mice that lack CD8^+^ DCs to define its potential involvement in Flagrp170-gp100 vaccination-induced antitumor immune response. We showed that the lack of CD8^+^ DCs reduced the proliferation and activation of adoptively transferred gp100-specific CD90.1^+^ Pmel cells following Flagrp170-gp100 immunization (Figure 5B). Further, the depletion of CD8^+^ DCs in Batf3^−/−^ mice abrogated the therapeutic activity of the Flagrp170-gp100 vaccine (Figure 5C), which was associated with sharply decreased levels of tumor-infiltrating IFN-γ^+^CD8^+^ T cells (Figure 5D), suggesting that the Flagrp170 chaperone vaccine-induced antitumor immune response is mediated by CD8^+^ DCs.

### 3.6. Flagrp170 Chaperone Vaccine Potentiates Tumor Response to Immune Checkpoint Inhibitor

Given the ability of the Flagrp170 chaperone vaccine to expand tumor-reactive T cells and immunologically remodel the tumor sites, we sought to examine its potential use for improving tumor responses to ICIs. C57BL/6 mice established with B16 melanoma were treated with the Flagrp170-gp100 complex, anti-PD-1 antibodies, or a combination. We showed that, while anti-PD-1 therapy alone failed to inhibit tumor progression, a combinatorial treatment resulted in the highly potent suppression of tumor growth (Figure 6A), which corelated with significantly prolonged animal survival (Figure 6B). Strikingly, approximately 60% of mice receiving combination therapy remained tumor-free (Figure 6B), suggesting a curative effect of the Flagrp170-gp100 complex vaccine in combination with anti-PD-1 therapy. These tumor-free mice were re-challenged with B16 tumors 90 days later and none of these mice developed tumors (Figure 6C), suggesting that a robust memory response has been established following the combination therapy.

## 4. Discussion

Although molecular chaperones or stress proteins are mostly involved in protein homeostasis [31], their unique capacity to shuttle client proteins and facilitate antigen cross-presentation have allowed them to be used in many vaccines or immunotherapeutic applications [32]. We have developed a chaperoning technology built on the superior protein antigen-holding feature of large chaperone molecules (Hsp110, Grp170) to construct synthetic chaperone complex vaccines targeting specific tumor antigens (e.g., Gp100) [14]. In addition to the safety profile and patient responses demonstrated in a phase I clinical trial [16], this recombinant chaperone vaccine approach offers unique advantages, including no requirement for surgical tumor specimen, unlimited quantities of off-the-shelf vaccines with uniformity, broad applicability, and easy immunomonitoring using defined antigens. Considering that coupling both an antigenic signal and immunostimulatory ‘danger’ molecule in the same vaccine delivery cargo is crucial for efficient immune activation [33,34], we have strategically constructed a new chimeric chaperone, i.e., Flagrp170, by fusing the chaperoning competent domain of Grp170 with a defined NF-κB-activating microbial element [35]. In the current study, we show that Flagrp170, as a new antigen-targeted vaccine platform, exhibits superior immunostimulatory activities compared to the unmodified parental chaperone molecule, which supports its potential use to develop a new-generation chaperone complex vaccine for clinical testing.

It has been documented that engaging innate pattern recognition signaling and subsequent NF-κB activation in APCs strongly promotes the immunostimulatory presentation of self/tumor antigens [23,24]. Our studies show that the chimeric Flagrp170 not only retains the ability to hold protein clients and interact with DCs, but also acquires a significantly enhanced capacity to stimulate TLR5-dependent NF-κB activation in DCs in comparison with the unmodified chaperone molecule. This is supported by our results from multiple experimental models, including the immunoblotting analysis of P65 phosphorylation and phosphorylation-triggered IκB degradation, the NF-κB reporter assay, and the TLR5-dependent activation of DCs and consequent T cell priming. Although the intracellular NLRC4 inflammasome can also serve as a receptor recognizing flagellin, NLRC4 deficiency does not appear to impact Flagrp170-induced cytokine production by DCs, whereas a lack of TLR5 abrogates its immunostimulatory activity. It is possible that the superiority of Flagrp170 may be due to its unique capacity to facilitate antigen cross-presentation via an interaction with endocytic receptor(s), while concurrently engaging TLR5-mediated immunostimulatory signaling cascade in APCs. However, additional studies are necessary to better understand the relative contributions of these two pattern recognition receptors to the immunogenicity of the Flagrp170–antigen complex in vivo, as well as the Flagrp170 action at molecular levels during its interaction with both endocytic and signaling receptors.

The Flagrp170 capable of the concurrent delivery of antigen targets, with a pathogen-associated molecular pattern for the optimal activation of DCs, profoundly enhances the cross-presentation of tumor antigens and the resultant antigen-specific T cell responses. Using mouse melanoma and breast cancer models as well as an experimental metastasis model, we have provided compelling preclinical evidence supporting the superior antitumor potency of the Flagrp170-based chaperone complex vaccine targeting gp100 or the Her2/neu antigen when compared to a chaperone vaccine similarly prepared with unmodified Grp170. The improved tumor control by the therapeutic Flagrp170 vaccine treatment is associated with a systemic generation and/or expansion of antigen-specific T cells, as well as an elevated tumor infiltration by T cells carrying an ‘activated’ effector phenotype. Our study also points to the important role of CD8^+^ DCs or classical type 1 DCs (cDC1) in directing and coordinating Flagrp170-triggered T cell response. CD8^+^ DCs or cDC1 cells are specialized in cross-presentation, or the presentation of the exogenous antigen on MHC class I, to induce naive CD8^+^ T cells to acquire CTL effector functions to respond to intracellular pathogens and tumors [36,37]. Our bilateral injection model shows that CD8^+^ DCs in the draining lymph nodes rapidly acquire a significantly enhanced maturation and activation phenotype upon the injection of Flagrp170 in comparison with the unmodified chaperone, suggesting that the embedded pathogen-associated molecular pattern provides the molecular chaperone with the ability to engage CD8^+^ DCs more efficiently. This is further supported by our studies involving Baf3-deficient mice that have a defect in the development of CD8^+^ DCs or cDC1 cells. The loss of CD8^+^ DCs in these mice abrogates the generation and activation of melanoma antigen gp100-specific CD8^+^ T cells and the consequent antitumor responses augmented by the Flagrp170-gp100 complex vaccine.

Despite the encouraging responses in certain patient populations undergoing immune checkpoint inhibition therapy, the response rate is low in most patients with advanced diseases. A poor responsiveness to immunotherapy (e.g., ICIs) results from a number of factors, including the loss of tumor antigen or MHC expression in cancer cells, immunosuppressive cells, and cytokines in the tumor microenvironment, a lack of pre-existing tumor-reactive T cells, or T cell exhaustion, which collectively contribute to an immunologically ‘*cold*’ tumor. Another major finding in the current study is that the Flagrp170-based chaperone complex vaccine can mobilize T cells effectively, resulting in high levels of immune infiltration in the tumor compartment and a remodeling of tumor immune niches, evidenced by the significant elevation of a gene signature linked to Th1/Tc1 immunity (e.g., IFN-γ, Granzyme B, IL-12, IL-15) or a T cell inflamed phenotype that can predict patient response to ICIs [8,9,10,11]. This enhanced immune activation status in the tumor sites also coincides with an upregulation of immune checkpoint molecules such as PD-1 and PD-L2. This observation is consistent with the previous report that an active T cell response in tumors is often associated with an elevation of immune checkpoint molecules, which may be indirect markers of activated CD8^+^ tumor-infiltrating lymphocytes [6,38]. Strikingly, the Flagrp170 chaperone complex vaccine greatly potentiates the responsiveness of inherently therapy-resistant B16 melanoma to anti-PD-1 antibodies, resulting in 60% of tumor-bearing mice being cured by the combination therapy. Furthermore, these tumor-free mice have developed fully protective antitumor immunity and can resist a secondary challenge with the same tumors. While the B16 melanoma model is well recognized for its poor immunogenicity, genetically engineered tumor models can be used in future studies to further evaluate the therapeutic activity of Flagrp170. Given that ICIs are known to unleash the suppressive mechanisms that restrain T cell functions, capitalizing on Flagrp170-based chaperone vaccination to amplify pre-existing antitumor T cell responses and transform tumor immune niches may not only help overcome cancer resistance to ICIs, but also can work synergistically with ICIs to maximize or broaden the immunotherapeutic benefits.

## 5. Conclusions

Together, our studies support the potential application of the new chimeric Flagrp170 molecule to develop novel chaperone complex vaccines, which may be strategically used to heighten tumor-reactive T cell responses for increased cancer susceptibility to ICIs. Given the safety profile and T cell-stimulating activity of the recombinant chaperone vaccine targeting melanoma antigen gp100 already established in the phase I clinical trial, our findings may provide a direct path to translating this next-generation chaperone complex vaccine from the laboratory into the clinic.

## Figures and Tables

**Figure 1 vaccines-13-01096-f001:**
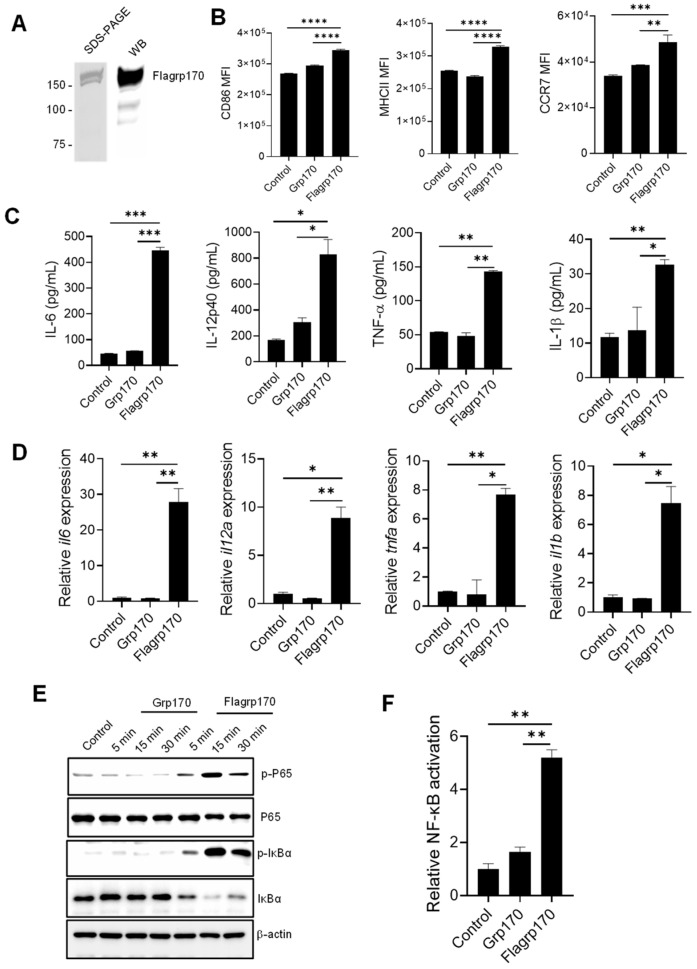
Flagrp170 protein is more effective than unmodified Grp170 in stimulating NF-κB-mediated activation of DCs. (**A**) Flagrp170 protein preparation was examined by gel staining and Western blot analysis. (**B**) Bone marrow-derived DCs were stimulated with Grp170 or Flagrp170 (20 µg/mL) overnight. The maturation and activation of DCs were assessed using flow cytometry. Expression of pro-inflammatory cytokines was detected with ELISA (**C**) and qPCR (**D**). (**E**) DCs were collected and analyzed for phosphorylation of P65 and IκBα through immunoblotting. (**F**) HEK293-TLR5 cells transfected with pGL3-NF-κB-Luc were treated with Flagrp170 or Grp170. NF-κB controlled luciferase activity was assayed 6 h later using a Glomax luminometer. Data representative of three independent experiments are shown. Student’s *t*-test was used. *, *p* < 0.05; **, *p* < 0.01; ***, *p* < 0.001; ****, *p* < 0.0001.

**Figure 2 vaccines-13-01096-f002:**
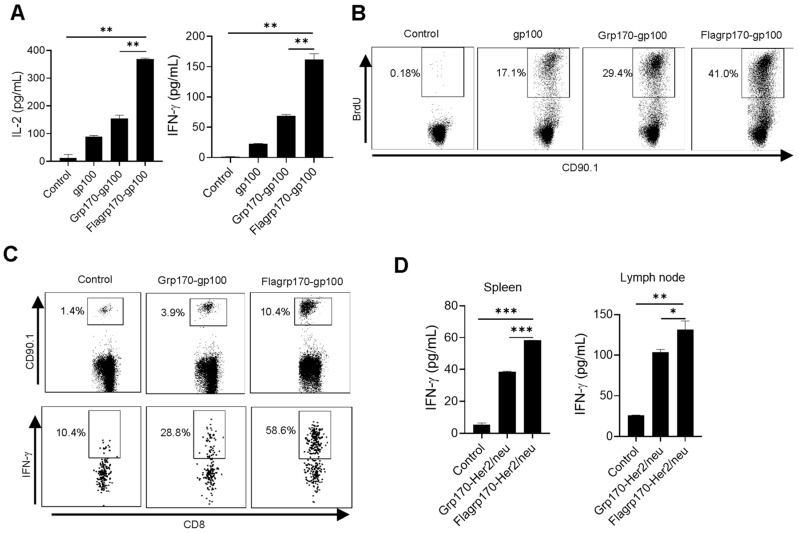
Flagrp170–antigen protein complex enhances DC-mediated T cell activation. DCs were pulsed by chaperone complexes and then co-cultured with CD8^+^ T cells purified from Pmel mice for 3 days in the presence of BrdU. IL-2 and IFN-γ production was assessed using ELISA (**A**) and BrdU incorporation was examined using flow cytometry (**B**). Pmel T cells were transferred into recipient mice (*n* = 5) through tail vein injection, which were immunized with Grp170-gp100 or Flagrp170-gp100 complexes the next day. Splenocytes were stained for CD90.1 and IFN-γ 3 days later to analyze Pmel cell proliferation and activation (**C**). C57BL/6 mice (*n* = 5) were immunized twice at weekly intervals with Grp170-Her2/neu or Flagrp170-Her2/neu complexes. Splenocytes or lymph nodes were collected 5 days later and stimulated with Her2/neu protein, followed by ELISA assays for IFN-γ production (**D**). Data representative of three independent experiments are shown. Student’s *t*-test was used. *, *p* < 0.05; **, *p* < 0.01; ***, *p* < 0.001.

**Figure 3 vaccines-13-01096-f003:**
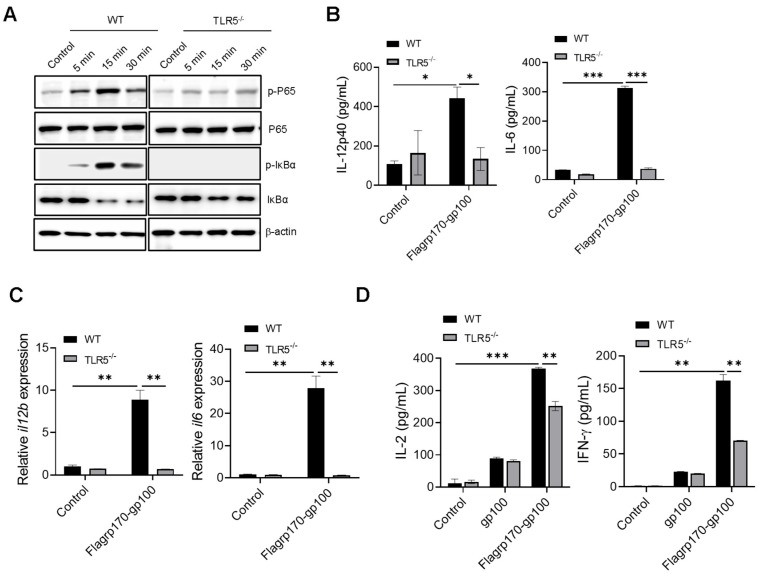
Enhanced activation of DCs by Flagrp170 depends on TLR5-mediated NF-κB signaling. Bone marrow-derived DCs from WT or TLR5^−/−^ mice were stimulated with Flagrp170 for indicated times and analyzed through immunoblotting for phosphorylation of P65 and IκBα (**A**). Following Flagrp170 treatment overnight, the expression of IL-12p40 or IL-6 by DCs was examined using ELISA (**B**) and qRT-PCR (**C**). WT or TLR5^−/−^ DCs were pulsed with Flargp170-gp100 complex and cultured with CD8^+^ Pmel cells for 3 days. T cell activation was examined using ELISA assays for IL-2 and IFN-γ production (**D**). Data representative of three independent experiments are shown. Student’s *t*-test was used. *, *p* < 0.05; **, *p* < 0.01; ***, *p* < 0.001.

**Figure 4 vaccines-13-01096-f004:**
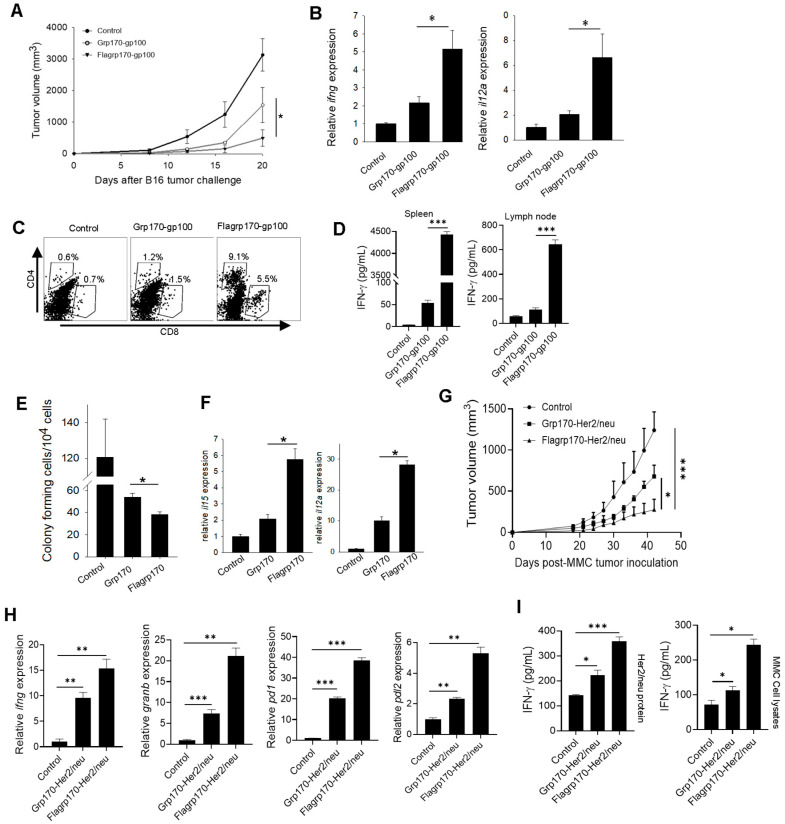
Flagrp170-based chaperone vaccine improves tumor inhibition and immune remodeling of tumor microenvironment. C57BL/6 mice bearing B16 tumors (*n* = 5) were immunized with Grp170-gp100 complex or Flagrp170-gp100 complex for three doses at 3-day intervals or were left untreated (**A**). Transcription of *il12a* and *ifng* genes in the tumors was examined through qRT-PCR (**B**). Tumor infiltration by T cells was assessed using flow cytometry (**C**). Splenocytes or lymph node cells were stimulated by gp100_25–33_ peptide, followed by ELISA assays for IFN-γ production (**D**). Experimental lung metastases were established through *i.v.* injection of B16 tumor cells, followed by treatment with chaperone vaccines (*n* = 5). Lungs were collected and prepared in single cell suspensions for clonogenic assays (**E**). The transcriptional levels of *il15* and *il12a* genes in the tumors were determined through qRT-PCR (**F**). FVBN202 mice established with MMC mammary tumors (*n* = 5) received immunization with chaperone-Her2/neu protein complexes (**G**). Tumor tissues were analyzed for changes in immune-related genes including *ifng*, *granb*, *pd1*, and *pdl2* (**H**). Splenocytes were stimulated with MMC tumor cell lysates or Her2/neu protein and then examined for IFN-γ levels in the culture media (**I**). Data representative of three independent experiments are shown. Student’s *t*-test and the two-way ANOVA test were used. *, *p* < 0.05; **, *p* < 0.01; ***, *p* < 0.001.

**Figure 5 vaccines-13-01096-f005:**
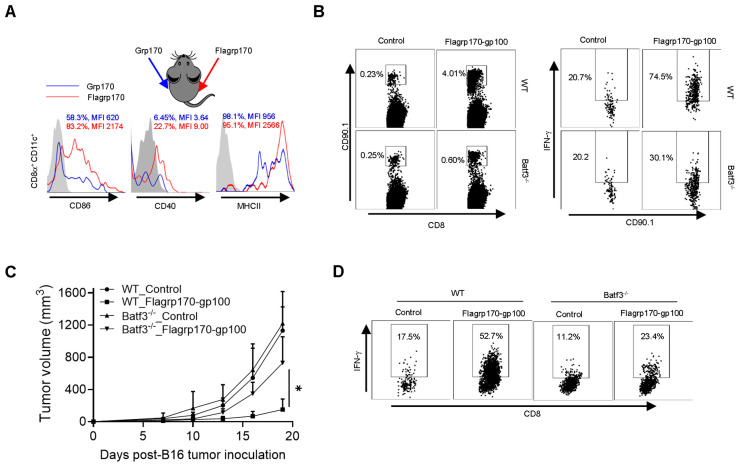
CD8^+^ DCs are required for therapeutic activity of Flagrp170-based chaperone vaccine. (**A**) Flagrp170 or Grp170 protein were injected into the opposite sides of the same mouse. Draining lymph nodes from each site were collected 16 h later and subjected to flow cytometry analysis for the status of CD8^+^CD11c^+^ DCs. The percentage of cells with positive staining for CD86, CD40, or MHC II and mean fluorescence intensity (MFI) are shown. (**B**) WT or Batf3^−/−^ recipient mice were transferred with Pmel T cells and immunized with Flagrp170-gp100 protein complex the next day. Splenocytes were harvested 3 days later to analyze proliferation and activation of Pmel cells. (**C**) WT or Batf3^−/−^ mice (*n* = 5) established with B16 tumors were treated with Flagrp170-gp100 complex vaccine three times. (**D**) Immune infiltration of tumors was examined using flow cytometry. Data representative of three independent experiments are shown. The two-way ANOVA test was used. *, *p* < 0.05.

**Figure 6 vaccines-13-01096-f006:**
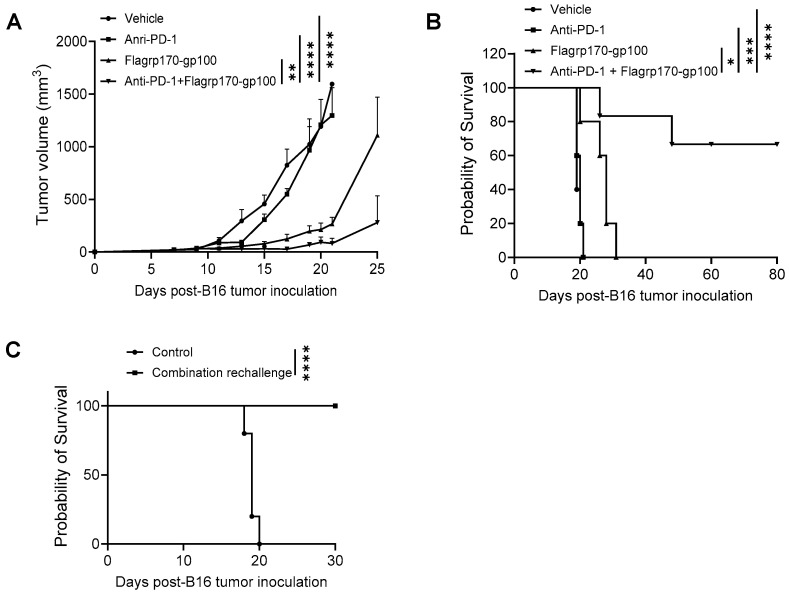
Flagrp170-based chaperone vaccine potentiates tumor response to anti-PD-1 therapy. C57BL/6 mice with established B16 tumors were treated with Flagrp170-gp100 complex, anti-PD-1 antibodies, or a combination on day 5, day 8, and day 11 post tumor inoculation. Tumor growth (**A**) and mouse survival (**B**) were monitored. (**C**) Tumor-free mice from the combination therapy group were rechallenged with live B16 tumor cells. Mice without treatment were used as controls. Data representative of three independent experiments is shown. The two-way ANOVA test and the log-rank test were used. *, *p* < 0.05; **, *p* < 0.01; ***, *p*<0.001; ****, *p* < 0.0001.

## Data Availability

Data will be made available on request.

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
