# Peer review of "A Novel Chaperone-Based Cancer Vaccination Enhances Immunotherapeutic Responsiveness Through T Cell Amplification and Tumor Immune Remodeling"

_vaccines, 2025, doi:10.3390/vaccines13111096_

Round 1

Reviewer 1 Report

Comments and Suggestions for Authors

In this study, Cheng et al. discussed the construction of the chimeric molecule, Flagrp170, which fuses the antigen-presenting capability of the chaperone Grp170 with the TLR5 agonist activity of flagellin. The study not only demonstrates the superior efficacy of Flagrp170 but also delves deeply into its mechanism of action, including its dependence on the TLR5-NF-κB signaling pathway and the critical role of CD8+ DCs. In addition, the research shows a synergistic effect, and even cures, when combining the Flagrp170 vaccine with anti-PD-1 therapy in the refractory B16 melanoma model, while also inducing immune memory. Overall, this manuscript presents a promising and well-validated vaccine delivery platform. However, there are some flaws is the experiment designs:

Major concerns: 

  1. A critical missing control is the use of the flagellin-derived peptide alone, simply mixed with Grp170 and the antigen. This control is essential to prove the unique advantage of the physical fusion strategy. The current results show Flagrp170 is better than unmodified Grp170, but they cannot fully rule out that a "simple mixture" might achieve similar effects. The fusion protein may offer better stability, targeting, or synergy, but this needs comparison with a "mixture group" to be confirmed.
  2. The article mentions that Grp170 preferentially binds to APCs via its receptors, but it does not investigate whether Flagrp170 retains this interaction and what role it plays. The superiority of Flagrp170 might stem from the combined effects of TLR5 activation and Grp170-receptor-mediated targeting/internalization, but the latter is not explored.
  3. The study focuses on IFN-γ production and BrdU incorporation (proliferation) but lacks a deeper characterization of T-cell differentiation states (e.g., effector memory vs. central memory T cells) and exhaustion markers (e.g., PD-1, TIM-3, LAG-3). This is particularly relevant for the combination therapy with anti-PD-1, where analyzing how the vaccine alters the exhaustion status of tumor-infiltrating T cells would be more compelling.
  4. The focus on DCs and T cells is justified. However, the tumor immune microenvironment includes various immunosuppressive cells, such as regulatory T cells (Tregs) and myeloid-derived suppressor cells (MDSCs). While the study shows a shift towards a Th1/Tc1 phenotype, it does not assess whether the Flagrp170 vaccine can suppress the infiltration or function of these suppressive populations.
  5. While two models are used, they are both transplantable tumor models, whose microenvironment may not fully recapitulate the complexity of human spontaneous cancers. Validation in genetically engineered models that more closely mimic human cancer genetics would enhance the clinical relevance. Furthermore, while the B16 model is challenging, it is still a highly immunogenic model; its efficacy in less immunogenic "cold" tumors needs further validation. This should be elaborated on in the Discussion section(minor).

Author Response

In this study, Cheng et al. discussed the construction of the chimeric molecule, Flagrp170, which fuses the antigen-presenting capability of the chaperone Grp170 with the TLR5 agonist activity of flagellin. The study not only demonstrates the superior efficacy of Flagrp170 but also delves deeply into its mechanism of action, including its dependence on the TLR5-NF-κB signaling pathway and the critical role of CD8+ DCs. In addition, the research shows a synergistic effect, and even cures, when combining the Flagrp170 vaccine with anti-PD-1 therapy in the refractory B16 melanoma model, while also inducing immune memory. Overall, this manuscript presents a promising and well-validated vaccine delivery platform. However, there are some flaws is the experiment designs:

We thank the reviewer for the time involved in reviewing the manuscript and appreciate the reviewer’s constructive and insightful comments. Below are our responses to the concerns raised by the reviewer.

Major concerns: 

Comments 1. A critical missing control is the use of the flagellin-derived peptide alone, simply mixed with Grp170 and the antigen. This control is essential to prove the unique advantage of the physical fusion strategy. The current results show Flagrp170 is better than unmodified Grp170, but they cannot fully rule out that a "simple mixture" might achieve similar effects. The fusion protein may offer better stability, targeting, or synergy, but this needs comparison with a "mixture group" to be confirmed.

Response 1: We appreciate the suggested experimental design involving use of the “simple mixture” control. We would also like to point out that the current study is focused on testing of a chimeric chaperone molecule that we engineered as a novel vaccine platform to deliver tumor antigens. In our previous work (Yu, et al., Cancer Research 2013. 73(7): p. 2093-103), we have demonstrated that Flagrp170 is superior to flagellin when administrated to the tumor lesions to induce antitumor immunity, even though flagellin is inherently immuostimulatory as a PAMP. We have made a distinct finding that Flagrp170 is significantly more effective than flagellin in generating tumor or antigen-specific CTL response. This phenomenon has been attributed to antigen-capturing/shuttling/presenting abilities of Flagrp170, which retains the chaperoning capacity of the parental chaperone Grp170. Indeed, our early research has shown that antigen complexed to a chaperone molecule is critical for induction of antigen-specific T cell response, because simply mixing antigen with chaperone molecule fails to generate an effective antitumor immune response (Wang et al., Cancer Research 2003. 63(10): p. 2553-60.). Additionally, it has been well recognized that coupling both an antigenic signal and immunostimulatory ‘danger’ molecule in the same vaccine delivery cargo is crucial for efficient immune activation (Blander, J.M. Nature, 2006. 440(7085): p. 808-12; Burgdorf, S. Nat Immunol, 2008. 9(5): p. 558-66.). While the ‘mixture’ control is not included in our study, we have provided the compelling evidence showing that Flagrp170 as the next generation chaperone-based immune modulator is significantly more potent than its parental chaperone molecule Grp170. Given that the unmodified chaperone molecule (i.e., Hsp110, the cytosolic homolog of Grp170) has been tested in melanoma patients in a phase I trial, our current study together with our previous publications strongly supports the potential use of Flagrp170 for developing new and more potent chaperone-based immunotherapies including cancer vaccines.

Comments 2. The article mentions that Grp170 preferentially binds to APCs via its receptors, but it does not investigate whether Flagrp170 retains this interaction and what role it plays. The superiority of Flagrp170 might stem from the combined effects of TLR5 activation and Grp170-receptor-mediated targeting/internalization, but the latter is not explored.

Response 2: We agree with the reviewer that the superiority of Flagrp170 may be due to its unique capacity to facilitate antigen cross-presentation via interaction with endocytic receptor(s) while concurrently engaging TLR5-mediated immunostimulatory signaling cascade in APCs. We have shown in this study that Flagrp170 protein can binds to APC in a dose-dependent manner, suggesting its potential interaction with its putative receptors on the surface of APC. Although multiple scavenger receptors have been identified as the binding structures for chaperone molecules on APCs, it remains unclear which receptor(s) can mediate the internalization and uptake of a chaperone-complex vaccine, and more importantly are responsible for inducing consequent antitumor immune responses. These studies are necessary to better understand the Flagrp170 action as an immune modulator at molecular level. A brief discussion has been included in our revised manuscript.

Comments 3. The study focuses on IFN-γ production and BrdU incorporation (proliferation) but lacks a deeper characterization of T-cell differentiation states (e.g., effector memory vs. central memory T cells) and exhaustion markers (e.g., PD-1, TIM-3, LAG-3). This is particularly relevant for the combination therapy with anti-PD-1, where analyzing how the vaccine alters the exhaustion status of tumor-infiltrating T cells would be more compelling.

Response 3: We thank the expert reviewer for the intriguing idea of potential alteration of the exhaustion status of TILs by the Flagrp170 vaccine. In this study, we show that immunization with the Flagrp170-antigen protein complex induces upregulation of immune checkpoint genes (e.g., pd1 and pdl2) in the tumor sites (Figure 4H), suggesting that immune activation by Flagrp170 chaperone vaccine is likely to trigger the immune counterbalancing mechanism in the activated T cells. Nonetheless, the increased enrichment of T cells in the tumor sites and their enhanced activated phenotype, evidenced by the elevated expression of IFN-g, are crucial for the Flagrp170 chaperone vaccine-induced antitumor immunity and sensitizing tumor response to immune checkpoint inhibition therapy. A future study can be performed to characterize the phenotype of T cells in details using more exhaustion and/or differentiation markers.

Comments 4. The focus on DCs and T cells is justified. However, the tumor immune microenvironment includes various immunosuppressive cells, such as regulatory T cells (Tregs) and myeloid-derived suppressor cells (MDSCs). While the study shows a shift towards a Th1/Tc1 phenotype, it does not assess whether the Flagrp170 vaccine can suppress the infiltration or function of these suppressive populations.

Response 4: Thank you for your thoughtful consideration. While we did not directly examine Treg and MDSCs in the current study, we have analyzed the transcriptional expression of foxp3 gene in the tumor tissues using qRT-PCR. We show that the foxp3 mRNA is elevated in tumors following immunization with Flagrp170 chaperone vaccine (Supplementary Fig. 4D), suggesting that Treg may be mobilized in responding to the Flagrp170-induced immune activation. This is not unexpected given the potent immune augmentation by the Flagrp170 chaperone vaccine and may indicate an immune homeostatic mechanism being engaged, which is also consistent with the PD-1 result. We have included this data in the Supplementary Figures of our revised manuscript. We agree it is worth conducting additional studies to investigate other immunosuppressive cells in the setting of Flagrp170 chaperone vaccination in the future study.

Comments 5. While two models are used, they are both transplantable tumor models, whose microenvironment may not fully recapitulate the complexity of human spontaneous cancers. Validation in genetically engineered models that more closely mimic human cancer genetics would enhance the clinical relevance. Furthermore, while the B16 model is challenging, it is still a highly immunogenic model; its efficacy in less immunogenic "cold" tumors needs further validation. This should be elaborated on in the Discussion section(minor).

Response 5: The B16 melanoma is considered poorly immunogenic or immunologically “cold” because established B16 tumors do not respond well to PD-1 Ab treatment, as shown in the current study. Additionally, our published work has demonstrated that immunization with irradiated B16 tumor cells cannot provide protective immunity against challenge with viable B16 tumor cells (Wang et al., Journal of Immunology, 2006. 177 (3), 1543-1551; Cancer Research, 2007. 67 (10), 4996-5002). However, we agree that genetically engineered tumor models can be used to further evaluate the therapeutic activity of Flagrp170 chaperone vaccine complex to support its translational potential. A brief discussion has been added to the discussion section of our revised manuscript.

Reviewer 2 Report

Comments and Suggestions for Authors

Cheng, X. et al outlined a series of experiments for evaluating the effects of the Flagrp170 fusion chaperone protein, which was previously established by this group, as a vaccine alone or combined with anti-PD1 in treating murine B16 melanoma and breast cancer. The impacts of Flagrp170 on BM-derived DC and tumor antigen-specific systemic and intra-tumoral CD8 T cells and the role of host cDC1 in the Flagrp170 vaccination were examined. Overall, the data are clear with proper interpretations for supporting the conclusions.  

Author Response

Cheng, X. et al outlined a series of experiments for evaluating the effects of the Flagrp170 fusion chaperone protein, which was previously established by this group, as a vaccine alone or combined with anti-PD1 in treating murine B16 melanoma and breast cancer. The impacts of Flagrp170 on BM-derived DC and tumor antigen-specific systemic and intra-tumoral CD8 T cells and the role of host cDC1 in the Flagrp170 vaccination were examined. Overall, the data are clear with proper interpretations for supporting the conclusions.  

Response: We appreciate your positive comments on our study.

Reviewer 3 Report

Comments and Suggestions for Authors

The study presents a chimeric vaccine platform (Flagrp170) that couples antigen chaperoning with innate activation to enhance dendritic-cell cross-presentation and anti-tumor CD8⁺ T-cell responses. Using murine melanoma models and antigen-specific readouts, the authors show improved T-cell priming, tumor control, and synergy with PD-1 blockade. The work is timely and promising; with tighter methodological reporting, clearer mechanistic evidence for cDC1 involvement, and more consistent figure documentation, the manuscript would be substantially strengthened.

Suggestions for improvement

1- Rigor and statistics.
In each figure legend, report biological n (mice per group) and whether points represent biological vs technical replicates; name the exact test used, state assumption checks, the multiple comparison method (Tukey/Sidak or FDR), exact/adjusted P-values, and effect sizes. 

2- Flow cytometry identity and controls.
Provide a complete gating strategy. To support cDC1 assignment, include subset-defining markers. If adding markers is not feasible, soften claims to avoid over interpreting subset identity.

3- Protein quality and endotoxin burden.
Report endotoxin levels for each in vivo lot and include purity/aggregate assessments. Add specificity controls (e.g., mutant/heat-inactivated flagellin domain) to confirm that DC activation arises from the intended construct rather than residual contaminants.

Author Response

The study presents a chimeric vaccine platform (Flagrp170) that couples antigen chaperoning with innate activation to enhance dendritic-cell cross-presentation and anti-tumor CD8⁺ T-cell responses. Using murine melanoma models and antigen-specific readouts, the authors show improved T-cell priming, tumor control, and synergy with PD-1 blockade. The work is timely and promising; with tighter methodological reporting, clearer mechanistic evidence for cDC1 involvement, and more consistent figure documentation, the manuscript would be substantially strengthened.

We thank you for your time in reviewing our manuscript and appreciate your comments.

Suggestions for improvement

Comments1: Rigor and statistics.
In each figure legend, report biological n (mice per group) and whether points represent biological vs technical replicates; name the exact test used, state assumption checks, the multiple comparison method (Tukey/Sidak or FDR), exact/adjusted P-values, and effect sizes. 

Response 1: Five mice per group were used and the data points were biological replicates. We have included these details in both method section and figure legends. We also added the details of test used in the method section.  

Comments 2: Flow cytometry identity and controls.
Provide a complete gating strategy. To support cDC1 assignment, include subset-defining markers. If adding markers is not feasible, soften claims to avoid over interpreting subset identity.

Response 2: As suggested, we added the complete gating strategy to the supplementary material. More details of cDC1 assignment have been included in figure legends.    

Comments 3: Protein quality and endotoxin burden.
Report endotoxin levels for each in vivo lot and include purity/aggregate assessments. Add specificity controls (e.g., mutant/heat-inactivated flagellin domain) to confirm that DC activation arises from the intended construct rather than residual contaminants.

Response 3: Baculovirus-insect cell system for protein expression typically produces protein with a very low endotoxin level, as previously described (Wang et al., Cancer Research 2003, 63 (10), 2553-2560; Wang et al., Journal of Immunology 2006, 177 (3), 1543-1551; Qian et al., Frontiers in Immunology 2023, 14, 111878). Endotoxin levels in the recombinant protein preparations are approximately 10 EU/mg protein. This information has been included in the revised manuscript. Although both proteins are prepared from the same expression system, Flagrp170 is significantly more efficient than the parental chaperone Grp170 in DC activation, as shown in the current work, suggesting that the potential contaminants in the protein preparations are not involved in immune stimulation.